# Coping Strategies during Childbirth Related to Cultural Identity: Companionship, Choice of Analgesia and Maternal Satisfaction

**DOI:** 10.3390/healthcare11121714

**Published:** 2023-06-12

**Authors:** Silvia Navarro-Prado, María Angustias Sánchez-Ojeda, Fernando Jesús Plaza del Pino, María Ángeles Vázquez-Sánchez, María Isabel Tovar-Gálvez, Nurimán Azirar-Mohamed

**Affiliations:** 1Department of Nursing, Faculty of Health Sciences of Melilla, University of Granada, 52017 Melilla, Spain; 2Department of Nursing, Faculty of Health Sciences, University of Almería, 04120 Almería, Spain; ferplaza@ual.es; 3Department of Nursing, Faculty of Health Sciences, University of Malaga, 29071 Malaga, Spain; mavazquez@uma.es; 4Department of Nursing, Faculty of Health Sciences of Ceuta, University of Granada, 51001 Ceuta, Spain; 5Centre for the Temporary Stay of Immigrants of the Autonomous City of Melilla, 52004 Melilla, Spain; nuriman.azirar@cruzroja.es

**Keywords:** childbirth, patient satisfaction, culture, epidural, companion

## Abstract

Childbirth is a biological process and how it is experienced and managed is influenced by numerous factors, among them, socio-cultural or health care received. Objective: The objective of this study is to ascertain whether cultural factors influence the way in which women deal with childbirth through the treatment of pain, companionship and maternal satisfaction. Methods: This study is a non-experimental, quantitative, ex post facto, cross-sectional study of women who gave birth in a border town in southern Spain. The sample consisted of 249 women. Results: No relationship was found between cultural factors and the choice of epidural analgesia, alternative methods to alleviate pain, being accompanied nor maternal satisfaction. There was a significant relationship between the type of companionship and with maternal satisfaction. Conclusions: Cultural factors did not influence how women dealt with dilation and childbirth. Results found that the person accompanying the mother was important for increasing maternal satisfaction. The intercultural training of healthcare professionals is necessary.

## 1. Introduction

Childbirth is a biological process, but the diversity of cultures offers a multitude of ways of coping with or perceiving childbirth [1]. Childbirth is one of the most relevant experiences in women’s lives. This experience can be perceived as satisfactory or unsatisfactory depending on numerous factors, including the care received during childbirth [2,3]. Among the factors that modulate this perception of satisfaction is the pain that the women feel. There are internal and external factors that can modulate women’s coping with labour pain. Among the internal factors are the subjective perception of pain, the creation of expectations about pregnancy and childbirth, fear, anxiety, physical-emotional state and obstetric history. All these factors can be modulated by external circumstances such as the health and family care received, the accompaniment during childbirth, available methods in the health centre to alleviate pain and the cultural patterns in which women live. The way of coping with pain is mediated by sociocultural, psychological, genetic and demographic factors, and by the expectations created by mothers-to-be [2,4]. Undoubtedly, knowledge of the quality of health services in the area of the parturient also plays a major role in coping with pregnancy, childbirth and parenting. Historically, pain has been one of the major worries of women in labour, classified as severe, unbearable and exhausting, but at the same time, expressed differently depending on the woman’s characteristics [5,6,7,8].

This pain may become distressed when the expectations of the woman in labour are not satisfied, which leads to a decrease in maternal wellbeing that causes more insecurity in this very important experience in women’s lives. Among the complaints most frequently expressed by puerperal women when interviewed about their satisfaction with the care received during childbirth, they indicate that insufficient pain management and the impossibility of being accompanied throughout the delivery process are the factors that most reduce their satisfaction [3,6,7]. In contrast, when pain is treated appropriately, there is emotional support from relatives and health personnel, pregnancy planning and attending maternal education, the prevalence of satisfaction is greater. When women in labour cannot be accompanied by family, partners or friends, if the midwife offers personalised, close care and is sympathetic to the woman’s wishes, the experience tends to be perceived as being of higher quality and experienced positively. This situation is not frequent in our environment due to the workload of health personnel. This means that in many cases they do not have the necessary time to provide close care throughout the birth process [4,8].

Migratory movements have now become a global phenomenon that brings multicultural richness to society. Each culture has its own customs and rituals that add differential aspects that intermingle among the population [9,10]. Since the 1960s, women’s migration has been rising, accounting for 49–51% of all migration. This increase has been called feminisation of migratory movements, but it does not only refer to the increased number of women migrants, but also to the change in the way they migrate. Whereas women in the 1960s migrated with their families, husbands or in family groups, today they migrate more independently. Nowadays, women migrate alone to look for work and to be the sole breadwinners for their families in their countries of origin. The feminisation of these migratory movements that has taken place in recent years has made the treatment of birth one of the most highly demanded healthcare provisions [11,12]. Midwives commonly treat labouring women from all over the world whose culture differs considerably from their own. Therefore, it is essential to be sensitive to the characteristics of each culture [13,14,15], as this generates a different perspective on the way women experience childbirth, and therefore to provide quality health care, the training of nurses and midwives needs to have a multicultural approach [16,17,18,19].

Knowing the sociocultural characteristics of women before the birth can help us to understand whether pharmacological methods are accepted to relieve pain, the preference for alternative methods to just alleviate it, the acceptance of companionship during labour or whether, in contrast, culturally they usually give birth alone [16]. Furthermore, we must be aware of the cultural acceptance of facts that can be controversial depending on the society in which the birth takes place, such as exaggerated verbalisation, no contact with the male partner during dilation or continuous praying. It is vital to delve deeper into this subject to be able to provide suitable healthcare in line with cultural diversity, which allows for reducing the incidence of parturient women with a high degree of dissatisfaction during labour.

The study takes place in a border city in southern Spain (Melilla), which is one of the main entry routes for Africans arriving in Europe [20], using Spain as a transit country [21]. It is a city where four cultures coexist: Christians, Jews and gypsies of European origin and Muslims of Berber origin. The majority groups are Christians of European origin and Muslims of Berber origin [9]. The main language used in the city is Spanish, but in the last 20 years, the use of Tamazight (Berber language) has spread and is now the only language known by a large percentage of the Muslim population. This fact means that many people have a language barrier because, although it is widespread, Tamazight is not known by the rest of the communities [22].

Birth care in Melilla is characterised by its respectful practices throughout the entire process. Midwives inform and support women during pregnancy, childbirth and postpartum. Midwives offer prenatal education to pregnant women that will allow them to develop a Birth Plan that truly addresses their preferences, but for that purpose, it is essential that women are informed without cultural bias.

Although the progression of each birth cannot be predicted, the Birth Plan reflects, in writing, each woman’s preferences and wishes regarding the care of the birth process. This implies fluent communication between health personnel and pregnant women. Melilla has only one hospital where, although respectful birth care is a priority, there are limitations that midwives communicate to women so that the birth plan chosen is in accordance with the real situation in Melilla:-Pain management. Women are informed of the possibility of pharmacological and non-pharmacological analgesia. In the case of pharmacological analgesia, they are informed that, if they choose epidural analgesia, it is essential that they carry out the analyses contemplated in the pregnancy follow-up, in case they don’t, they may not be able to fulfill their wish if the delivery is accelerated and it is not possible to carry out the blood tests required by the anaesthesiology service. With regard to pain relief by non-pharmacological means, they must know it is impossible to request acupuncture because this service is not available at this hospital.-Birth environment. Pregnant women are informed of the possibility of adapting the environment to their wishes, controlling the light and the use of music therapy in an individualized way. Moreover, they are offered the possibility of being accompanied during the birth process. For this purpose, the possibility is also offered to the accompanying person to begin their work from the earliest stages of pregnancy, by encouraging them to attend childbirth preparation classes.-Birth positions: among the limitations of the Melilla Regional Hospital is that midwives cannot provide the women the possibility of relaxing in the pool or using the obstetric chair for vertical birth. In addition, ambulation in the delivery room is limited, so women are informed that if they wish to ambulate, they must stay longer in the Maternity and Infant Department. Other positions obtained by means of a ball and vertical rope are possible.-Fetal monitoring. To facilitate ambulation in the maternity ward, wireless foetal monitoring is offered to facilitate the mobility of the woman in labour.-Obstetric interventions. In this respect, there are no major limitations in comparison to other Spanish hospitals, consequently, there are no differences related to the information provided.-Skin-to-skin contact and breastfeeding. One of the characteristics of birth care in Melilla is the importance given to breastfeeding from the early moment of birth. In this respect, birth mothers are informed about breastfeeding support groups as well as about the options for inhibiting breastfeeding if, after receiving all the information, the woman decides not to breastfeed naturally. In addition, the Regional Hospital encourages skin-to-skin contact between healthy and pathological newborns, making this option available to mothers and fathers in the Neonatology and Maternity and Infant Departments.

Historically, it has been observed that women in Melilla tend to reject the option of relieving the pain of childbirth and being accompanied during this process, this option apparently is based on women’s cultural identity.

Therefore, the aim of this research is to find out whether cultural identity influences the way women cope with childbirth. To this purpose, the research aims to study whether culture influences women’s choice of method of pain management during labour, the choice of a birth companion during labour and delivery, and maternal satisfaction.

## 2. Materials and Methods

### 2.1. Study Design

It is a non-experimental, quantitative, ex post facto cross-sectional study. The variables studied are:Sociodemographic variables: Age, age she had during her last childbirth, cultural identity, marital status, level of studies, employment situation and nationality.Obstetric variables (from last childbirth experience): The presence of a companion during labor, type of companionship, use of epidural analgesia or another alternative method to alleviate pain during labor, and why she refused to use these resources.Variables related to the quality of care: The subscale related to personal satisfaction of the woman during the last childbirth she experience, belonging to the Mackey Childbirth Satisfaction Rating Scale (MCSRS), was used.

### 2.2. Participants

This study was performed on a group of women who were cared for during childbirth in a border city in the south of Spain. Non-probability sampling was performed for convenience giving a sample of 249 women who gave birth in this city between 2000 and 2022. It should be borne in mind that, despite the long period of time studied, the sample is representative because the city serves women from Morocco who come to the city only to be attended to during childbirth. In addition, another large percentage of women residing in Melilla have a language barrier. All these women were excluded because they were unable to complete the questionnaire.

### 2.3. Instrument

An “ad hoc” questionnaire was produced, made up of closed multiple-choice ques-tions based on sociodemographic and obstetric characteristics, described above. To measure the satisfaction of women with the experience of labour and birth, the Mackey Childbirth Satisfaction Rating Scale (MCSRS) [23,24] was used, adapted and translated into Spanish [22]. It is formed by 35 items grouped into five subscales that make reference to the woman (9 items), the partner (2 items), the baby (3 items), the midwife (9 items) and the obstetrician (8 items). It also includes a subscale for general evaluation of the experience (3 items). Each item is evaluated on a Likert 5-point scale that ranges from very dissatisfied (1) to very satisfied (5). The total scale score is achieved by adding up the values assigned to each item, so that a higher score is higher satisfaction.

### 2.4. Procedure

To disseminate the questionnaire, we requested the collaboration of the midwives who work in the maternity service of the Spanish border city and who had personal contact with women who had been cared for in that birth centre and were therefore reg-istered in the archives. In turn, the collaboration was also asked of the women so they also distributed the questionnaire among their contacts. Inclusion criteria were established that at the last birth the women understood Spanish and had given birth to a live newborn. Women who made errors when completing the questionnaire, such as unanswered questions, were excluded.

### 2.5. Data Analysis

Data for qualitative variables are shown as absolute frequency and percentage, and were analysed using a Chi-square test, or Fisher’s exact test. Quantitative variables are expressed as mean (standard deviation). In the case of quantitative variables, the fit to Normality was checked using a Kolmogorov Smirnov test; in case of non-adjustment, the homogeneity of the distribution in the different groups was compared using a Mann-Whitney test or a non-parametric ANOVA (Kruskal-Wallis H-test) [25].

### 2.6. Ethical Considerations

The present study was conducted in compliance with the ethical principles set out in the Declaration of Helsinki. In addition, women participated voluntarily having signed an informed consent form. The confidentiality of the data and the anonymity of the par-ticipants were preserved at all times. This article was approved by the ethics committee of Comarcal Hospital of Melilla (Spain) with the registration number 201800500007736 on 14 September 2018.

## 3. Results

The age of the participants was 40.80 (9.42), the age they had during their last childbirth was 32.96 (4.88). The sociodemographic data can be consulted in Table 1.

It was also analysed if culture influences the method chosen by the woman in labor to treat pain. Table 2 shows the results obtained in relation to cultural identity and its association with the choice of epidural analgesia. The analysis shows that there are no significant differences between cultural identity and the decision to use epidural analgesia and cultural identity and the alternative methods to alleviate pain. Nevertheless, on analysing the reasons for rejecting epidural anaesthesia, a statistically significant variation is observed between the different cultural identities and the various reasons for rejecting this type of analgesia (*p* < 0.05). The “Other religions” group shows a higher proportion of women who preferred to experience natural childbirth. No significant differences were observed between cultural identity and the alternative methods used to alleviate labor pain. The Bobath ball and walking were the most widely used.

The presence of support during birth was not influenced by cultural identity, ac-cording to the test Fisher’s exact test (Table 3).

The relationship between cultural identity and the type of companionship during labour was studied using Fisher’s exact test, indicating that religion did influence the type of companionship during labour. Agnostic or Christian women were the least accompanied during childbirth by their partners, while Muslim women were accompanied by other family members (Table 3).

Finally, the results obtained to find out the personal satisfaction of the parturients ac-cording to cultural identity are shown in Table 4. Firstly, a score of the Adapted Mackey scale of 4.24 (0.77) was obtained. To do this, the Kruskal-Wallis H test was used, where it is observed that there is no significant relationship between both variables. Table 4 shows, in general terms, that the women who took part in this study had a high degree of satisfaction in relation to the experience of their last birth.

No significant differences in satisfaction are found between the different cultural identities, while women who gave birth accompanied were more satisfied with their experience.

Table 5 shows that the presence of companionship during labour did influence the degree of maternal satisfaction, corroborated with the non-parametric Mann-Whitney U test, which reports that there is a statistically significant variation between both variables.

Those women who did not have companionship during labour had greater dissatisfaction, those who had a figure for support showed greater satisfaction. Specifically, women who had the support of a loved one stated that they perceived higher satisfaction compared with those who did not have support.

## 4. Discussion

Cultural identity and its possible influence on pregnancy, childbirth and the postpartum period is one of the concerns in relation to maternal health care. The main objective of this research is to find out whether cultural identity influences women’s approach to childbirth. The main objective of this research is to find out whether cultural identity influences women’s choice of method of pain management in childbirth, choice of birth companion and maternal satisfaction.

This study revealed that most women decided to use epidural analgesia during labour, which coincides with the results obtained in other studies that indicated a rising trend in the use of this type of analgesia in parturient women, as labour pain is one of the main concerns of women in labour [26,27]. In relation to the reason for rejecting the administration of epidural analgesia, it was observed in this study that the motive was largely the non-existence of this type of analgesia at the time of birth, a fact which conditions the choice by the parturient woman, as the use of epidural analgesia in the hospital in this city took longer than in other Spanish cities and hospitals, a fact that is coincident with other studies [4,14,28,29,30].

Accompaniment during childbirth is an important factor for women and has an impact on their satisfaction. This is consistent with the results obtained by Markosyan et al. in which the presence of a family member during labour and delivery was a factor that increased women’s satisfaction [31]. Cultural identity was decisive in the choice of a delivery companion, and Christian and Muslim women were accompanied more than Jewish and Hindu women. This differs from the findings of Fernandez-Carrasco [29], who reported that five times more Christian parturients were accompanied than Muslim parturients. Luque and Oliver [32] showed that accompaniment during childbirth was influenced by culture. In particular, North African immigrant women, mostly Muslim, did not want to be accompanied by their partners, as they considered it a purely female event and preferred to be accompanied by a woman. This finding does not coincide with what the present study found, as Muslim women were mostly accompanied by their partners, and only chose a female figure in second place.

It was observed that companionship was closely related to greater maternal satis-faction with the care received, findings that coincide with other studies [26,33,34,35]. Although in this study there was no relationship between culture and maternal satisfaction, it is very important to bear in mind the culture of the parturients in order to not influence on their preferences and care negatively [29]. According to these authors, identifying women´s satisfaction serves as a tool to increase the quality of health care. They report that the higher the satisfaction, the greater the probability of better obstetric outcomes.

### Limitations

This study had several limitations that reduced the number of women participating. As the study was carried out in a city bordering Morocco, there was a large percentage of women who only stayed in Spain to be attended to during childbirth, returning to their country of origin after being discharged from hospital. On the other hand, among the women who resided in Spain, a considerable percentage of them presented a language barrier and therefore could not be taken into account in the study. Finally, as the questionnaire was released on the internet, its completion required women to have an electronic device with internet connection. Furthermore, due to the socio-economic characteristics of the city, it was impossible to access the population group of women who did not have a telephone or e-mail.

## 5. Conclusions

Cultural identity does not influence the choice of epidural analgesia, but it has an impact on the reason given for refusing pain relief in labour and delivery. Similarly, cultural identity does not influence the choice of accompaniment, but it does have an impact on the person the woman chooses at the time of delivery. Maternal satisfaction was not influenced by cultural identity, but was related to the woman’s choice of companionship during labour and delivery.

Results indicate that, even though it is not a determining factor in the manner of management with birth, the cultural identity of the women of this study does condition certain aspects that have to be taken into account. As health professionals, we have to be aware of the reasons why women refuse pain relief during labour and delivery in order to provide them with accurate information to guide their decision. In addition, we also need to give women in labour the possibility to choose a significant person to accompany them during the process. In this way, the quality of care, and thus maternal satisfaction, is enhanced by taking these cultural aspects into account.

## Figures and Tables

**Table 1 healthcare-11-01714-t001:** Sociodemographic data.

Cultural or Religious Identity		N (%)
	Agnostic	24 (9.6%)
	Christian	168 (67.5%)
	Muslim	49 (19.7%)
	Other monotheistic religions	8 (3.2%)
Nationality	Spanish	240 (96.4%)
	Foreign	9 (3.6%)
Employment situation	Employed	223 (89.6%)
	Unemployed	13 (5.2%)
	Housewife	10 (4%)
	Student	3 (1.2%)
Level of studies	University/Professional training	210 (84.3%)
	Secondary	26 (10.4%)
	Primary	7 (2.8%)
	No studies	6 (2.4%)
Marital status	Married	184 (73.9%)
	Stable partner	45 (18.1%)
	Separated	10 (4%)
	Single	8 (3.2%)
	Widow	2 (0.8%)
Number of children	1 child	92 (36.9%)
	2 children	123 (49.4%)
	3 children	22 (8.8%)
	4 children	12 (4.8%)

**Table 2 healthcare-11-01714-t002:** Choice of epidural analgesia, reason for rejection and other alternative methods to alleviate pain, depending on culture.

	Cultural Identity					
	n (%)					*p*-Value
Choice of epidural analgesia						
	Agnostic (n = 24)	Christian (n = 168)	Muslim (n = 49)	Other (n = 8)	Total (n = 249)	
Yes	19 (79.2)	131 (78.9)	38 (77.6)	5 (62.5)	195 (78.3)	
No	5 (20.8)	35 (21.1)	11 (22.4)	3 (37.5)	54 (21.7)	0.735 *
Reason for rejection						
	Agnostic (n = 5)	Christian (n = 35)	Muslim (n = 11)	Other (n = 3)	Total (n = 54)	
No possibility	1 (20.0)	23 (65.7)	4 (36.4)	0	28 (51.9)	
Afraid of side effects	1 (20.0)	3 (8.6)	2 (18.2)	0	6 (11.1)	
Contraindicated due to suffering from a pathology	0	0	1 (9.1)	0	1 (1.9)	
Wanted to experience birth naturally	1 (20.0)	5 (14.3)	2 (18.2)	2 (66.7)	10 (18.5)	
Not necessary, as the dilation was very advanced	2 (40.0)	4 (11.4)	2 (18.2)	1 (33.3)	9 (16.7)	0.050 **
Other alternative methods to alleviate pain						
	Agnostic (n = 10)	Christian (n = 57)	Muslim (n = 17)	Other (n = 2)	Total (n = 86)	
Use of ball	7 (70.0)	36 (63.2)	9 (52.9)	1 (50)	53 (61.6)	
Walking	1 (10.0)	4 (7.0)	2 (11.8)	0	24 (27.9)	
Performing massages	2 (20.0)	16 (28.1)	5 (29.4)	1 (50.0)	7 (8.1)	
Music therapy	0	1 (1.8)	0	0	1 (1.2)	
Immersion in water	0	0	1 (5.9)	0	1 (1.2)	0.756 **

* Chi square. ** Fisher’s exact test.

**Table 3 healthcare-11-01714-t003:** Presence of support and type of companionship during labour.

	Cultural Identity					
	n (%)					*p*-Value
Were you accompanied in the last birth?						
	Agnostic (n = 24)	Christian (n = 168)	Muslim (n = 49)	Other monotheistic religions (n = 5)	Total (n = 246)	
Yes	22 (91.7)	143 (85.1)	41 (83.7)	3 (60.0)	209 (85.0)	
No	2 (8.3)	24 (14.9)	8 (16.3)	2 (40.0)	37 (15.0)	0.302 *
Who were you accompanied by?						
	Agnostic (n = 23)	Christian (n = 167)	Muslim (n = 49)	Other monotheistic religions (n = 5)	Total (n = 244)	
Nobody	2 (8.7)	25 (15.0)	8 (16.3)	2 (40.0)	37 (15.2)	
Partner	18 (78.3)	120 (71.9)	25 (51.0)	2 (40.0)	165 (67.6)	
Other relatives	2 (8.7)	19 (11.4)	15 (30.6)	1 (20.0)	37 (15.2)	
Other friendships	1 (4.3)	3 (1.8)	1 (2.0)	0	5 (2.0)	0.028 *

* Fisher’s exact test.

**Table 4 healthcare-11-01714-t004:** Relationship between the Woman’s Personal Satisfaction Subscale with the cultural identity and the presence of companionship.

	Cultural Identity					
	M (SD)					*p*-Value
Maternal personal Satisfaction						
	Agnostic (n = 24)	Christian (n = 168)	Muslim (n = 49)	Other monotheistic religions (n = 5)	Total (n = 246)	
	4.09 (0.94)	4.27 (0.73)	4.10 (0.85)	4.84 (0.08)	4.24 (0.77)	0.358 *

* Kruskal-Wallis test.

**Table 5 healthcare-11-01714-t005:** Relationship between satisfaction and accompaniment during childbirth.

	Accompaniment during Childbirth		
	M (SD)		*p*-Value
Maternal personal Satisfaction			
	Accompaniment in childbirth (n = 210)	Unaccompanied childbirth (n = 39)	
	4.32 (0.68)	3.82 (1.06)	0.006 *

* Mann–Whitney U test.

## Data Availability

The datasets used and/or analyzed during the current study are available from the corresponding author on reasonable request.

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
