# Peer review of "Coping Strategies during Childbirth Related to Cultural Identity: Companionship, Choice of Analgesia and Maternal Satisfaction"

_healthcare, 2023, doi:10.3390/healthcare11121714_

Round 1

Reviewer 1 Report (Previous Reviewer 2)

no comments

Author Response

Thank you very much for taking the time to review our manuscript.

Reviewer 2 Report (Previous Reviewer 3)

There is a very significant improvement to the manuscript. It is evident that researchers are invested in improvement. 

A few minor concerns:

-Line 34 - a single line cannot be a paragraph, it it needs to be integrated into line 36. I suggest the authors response be used "diversity of cultures that offer a multitude of ways of coping with or perceiving childbirth" as it is straight forward. 

-Line 213 - typo "it was them analysed" - perhaps "then"

The manuscript should be reviewed again for typos. 

The manuscript should be reviewed again for typos and grammatical errors. There are a lot less from first submission but yet a few still remain. 

Author Response

Thank you very much for taking the time to review our manuscript. Your comments have encouraged us to improve the manuscript and to correct the previous version.

We would like to inform both the Reviewers and the Editor that a native speaker has checked the new version of the manuscript.

This manuscript is a resubmission of an earlier submission. The following is a list of the peer review reports and author responses from that submission.

Round 1

Reviewer 1 Report

The manuscript is interesting, well written and scientifically sound. Overall, design of study is fine and data is solid. However, minor errors were found. Please see below some minor suggestion for improvement.

a) These reference on the cultural identity and pregnancy should be included https://www.jognn.org/article/S0884-2175(15)33239-1/fulltext and https://bmcpregnancychildbirth.biomedcentral.com/articles/10.1186/1471-2393-13-205  

b) There is a typo error in line 39 “mothers. [4,2].”

c) I suggest including the name of the city in lines 73-74

d) In lines 103-106 the sentence is confused and should be rephrased

e) Please include these supporting references for the employed Mann-Whitney and ANOVA (Kruskal-Wallis H-test) tests PMID: 34970247 and for  Kolmogorov Smirnov test PMID: 34784456

f) study limitations should be moved above conclusions

Author Response

Responses to reviewers. Manuscript ID- 2303815

First of all, we would like to thank the reviewers for their suggestions, which we believe were necessary and have helped to improve the quality of this study. All changes have been highlighted in red in the manuscript. We now proceed to respond to each of their suggestions:

REVIEWER 1

  1. These reference on the cultural identity and pregnancy should be included https://www.jognn.org/article/S0884-2175(15)33239-1/fulltext and https://bmcpregnancychildbirth.biomedcentral.com/articles/10.1186/1471-2393-13-205. The change has been included
  2. There is a typo error in line 39 “mothers. [4,2].” The change has been included. We have also detected this error in the discussion section and it has also been rectified.
  3. I suggest including the name of the city in lines 73-74. The change has been included
  4. In lines 103-106 the sentence is confused and should be rephrased. We have rewritten the sentence
  5. Please include these supporting references for the employed Mann-Whitney and ANOVA (Kruskal-Wallis H-test) tests PMID: 34970247 and for  Kolmogorov Smirnov test PMID: 34784456. The 1st suggested reference has been added, but the 2nd one has not been found.
  6. Study limitations should be moved above conclusions. The change has been included

Reviewer 2 Report

1.      Non-probability sampling is a non-random trial of subjective choice and statistical inference should not be used.

2.      One may not identify cultural identity only with religion

3.      Table 1 is Results. It is placed in Methods section.

4.      How many questionnaires were distributed and what percentage were answered?

5.       Quoted authors’ names do not correspond with numbers in References section.

6.      More than 50% of citations are not in English.

Author Response

Responses to reviewers. Manuscript ID- 2303815

First of all, we would like to thank the reviewers for their suggestions, which we believe were necessary and have helped to improve the quality of this study. All changes have been highlighted in red in the manuscript. We now proceed to respond to each of their suggestions:

REVIEWER 2:

  1. Non-probability sampling is a non-random trial of subjective choice and statistical inference should not be used. The non-probabilistic design prevents making population estimates, such as the percentage of Muslim women in the city of Melilla, or the total percentage of women who refuse epidural analgesia, but does not interfere with the possible associations found in the subjects in the sample.
  2. One may not identify cultural identity only with religion. Among the instructions given to the participants was a clarification in this regard. It was emphasised that they should self-identify according to the culture that surrounds the religions, taking into account the customs, rites and practices within each one of them.
  3. Table 1 is Results. It is placed in Methods section. The change has been included
  4. How many questionnaires were distributed and what percentage were answered? The sampling method used makes it impossible to obtain these data.
  5. Quoted authors’ names do not correspond with numbers in References section. Indeed, we have noted numerous errors in the bibliography. We erroneously included references in Spanish when in fact they were in English. We have also substituted some Spanish references for others that were used in the initial design of this study but were not included in the initial version sent to Healthcare.
  6. More than 50% of citations are not in English. Answered in suggestion 5

Reviewer 3 Report

This is a very interesting research topic and women's health is a blossoming field. More research is needed on women's maternity and postpartum experiences & challenges.

The article presented has major concerns including overall organization/structural challenges, grammar and diction errors, and is written in a causal English language. Demographic data needs to be shown (missing age). I can see some of the points the article is trying to make but at times the paragraphs are too wordy. The article needs refinement.  

·        References are both in English and Spanish – how is the journal going to handle the ethical and academic responsible for a reader to review the articles?

·        If a table is to be created about sociodemographic data all should be included – the table is missing: ages

·        A table needed to show the prevalence rates of obstetric valuable in the same way the authors completed one for sociodemographic data

·        Researchers fail to state their hypothesis

·        Authors bring up the point about cultural identity of the midwife and how that can be different than the mothers. This is a valid and crucial point in clinical practice of multiculturalism. However, in their article and research section no further point about the differences between midwife and patient culture is mentioned. It is just brough up once and left alone. Later in the article it discusses cultural identity but refers to the mothers. Authors need to remove the language about differences in culture between midwife & mother or address it more in the article. As it is currently written the differences of culture between midwife and mother is a stand alone irrelevant point.

·        Results section can be better organized by dividing it into subtitles of the result. As it is currently written it is one long paragraph and readers can be confused about the points concluded.

----

·        Missing from abstract = brief sentence describing population (e.g. N size, population, etc.)

·        Line 18 – “dealt with” has a negative connotation, neutral term is needed (e.g. managed, processed, etc.)

o   Authors make it a point to described childbirth as both positive and negative so a neutral term is needed

·        Line 19 – “the objective” incomplete phrase – perhaps “the objective of the study”

·        Line 21 – “it is” needs to be rephrased to – “this study is”

·        Line 22 – “was made” – researchers did not make N size, perhaps meant “selected”

·        Line 23 – “no relationship” is not capitalized which is inconsistent with prior words after headings

·        Line 25 – “however” is not needed

·        Line 26 – “cultural” is not capitalized which is inconsistent with prior words after headings

·        Line 27 – “Results found that” language is needed

·        Line 28 is most appropriate for conclusion/discussion section

·        Line 32 – The point of this sentences is unclear. Is culture the diversity?

·        Line 37 – “how this is face” – how what is faced: childbirth? Pain?

·        Line 41 – incorrect usage of “expressed” – perhaps authors mean “experience”

·        Line 44 – What stage?

·        Line 48 – Missing a period

·        Line 50 – This is not a new paragraph. This is a continuing idea from prior sentences.

·        Line 50-52 – This sentence cannot be understood as it is currently written and has no clear point. Perhaps authors are stating the prevalence of satisfaction is greater when pain is treated appropriately? Are the factors of appropriate pain management include emotional support, pregnancy planning, maternal education?

·        Line 53 introduces the idea of migration. Authors need a way to clearly divide the introduction with subheadings (childbirth, migration, etc.).

·        Line 55 – “feminisation” is word to describe submissive sexual practices (dominance, submission, kink). More appropriate word choice is needed.

·        Line 60- “greatly appreciate” – women do not appreciate multiculturalism in this context. Are authors trying to state the women in the childbirth process have benefits because of the intersections of cultures between midwife & mother to be?

·        Line 63 – “in short” wordiness, not needed. It retracts from authors argument

·        Line 64 – who is “us”? researchers, midwives, doctors, pain doctors, community as whole?

·        Line 64 – “or not” wordiness, not needed

·        Line 65 – “alleviate it” needs to be clearly written because it is an unclear reference, alleviate pain?

·        Line 66 – “or whether, in contrast, culturally they usually give birth alone” wordiness. Authors use the phrase “acceptance of companionship” which already insinuated labor alone or with company.

·        Line 63-67 – perhaps authors can benefit from the phrase “birth preferences/birth plan of mother influences by her culture?”

·        Line 67 – “we” who is we referring to?

·        Line 67 – 69 – confusing sentence. Are these facts or preferences selected by the mother due to culture? Authors are listing examples of these – exaggerated verbalization, no contact with male partner, continue praying?

·        Line 69 – missing comma

·        Line 70-72 is not a new paragraph. This is a concluding statement from prior paragraph.

·        Line 71-72 – do authors mean “allows the reduction of dissatisfaction of parturient women during labor”?

·        Primary languages spoken by mothers is missing (percentages of what mother spoke what language) in tables. This is attempted to be address in line 77-81 but authors are not clear.

·        Line 77-81: Authors do not need to justify the WHY behind the different languages spoken in the area. This sentence can be removed. Just the data is needed.

·        Line 73 – what is the name of the border city? What do locals call the location? How is it referred to?

·        Line 94 – can be moved to line 92 to make sentence clearer – perhaps obstetric values from last childbirth experience

·        Line 95 – incorrect tense – perhaps authors mean “experience”

·        Line 98 – no comma needed

·        Line 127 – incorrect spelling “es-tab-lished”

·        Line 129 – incorrect spelling “question-naire”

·        Line 127 – 130 – sentence needs to be restricted to clearly state inclusion and exclusion criteria. One sentence for inclusion. One sentence for exclusion

·        Line 130 – why were inconsistence answers excluded

o   This could be a major design flaw biasing the data. Researchers need to give a brief rationale for excluding inconsistence and what was the criteria for inconsistencies

·        Line 146 -  “firstly” is not needed. This is a study, researchers do not need to show step by step how analysis was completed.

·        Line 146 – who is “it”?

·        Line 150 – A briefer sentence of table 2 needed. If readers want more information they can seek out table 2

·        Line 148-149 – this is repeated from line 132-133. Authors need to list results not repeat what test was selected.

·        Line 157 – “p =.05” it is odd that researchers are using an “=” sign, perhaps “p<.05”?

·        Line 159 – “which explains this result” – explains what result?

·        Line 159 – Incorrect grammar, perhaps “no significant differences WERE observed”

·        Line 160 – the method of Bobath needs to be a separate sentence

·        Line 171 – “the results itemised in Table 3” language awkward, perhaps authors mean “see table 3”

·        Line 174 – t is lowercase, needs to capitalized

·        Table 4 – there is a comma in Agnostic (4,09)

·        Line 200 – “to this end” is casual filler language, can be removed

·        Line 202 – “reveals” needs to be passed tense

·        Line 206-209 can be combined with prior paragraph

·        Line 206-207 – incorrect grammar, perhaps “it was observed”

·        Line 206-209 – unclear what authors are trying to state regarding non-existence and motive. Authors did not examine motive.

·        Line 210 – “accompaniment during childbirth is an important factor for midwives”. This is a brand-new idea and it is unclear where this sentence is substantiated from the data. Perhaps authors mean it is important for mothers, not midwives?

·        Line 212 – how is Arbues Cultural identity decisive when choose a companion? Did researcher time mothers on how long it took them to choose a partner? Perhaps decisive is not the appropriate word choice?

·        Line 215 – 216 – this sentence is more appropriate for introduction section as it discusses prior literature.

·        Line 218 – “this fact” fact is not the appropriate word choice, this is a preference by mothers not fact. Perhaps researchers mean observation or preference?

·        Line 222 – use of the word “fact” - fact is not the appropriate word choice, Perhaps researchers mean observation or preference?

·        Line 225 – “according to these authors” which authors – the authors of this study or Fernandez-Carrasco et. Al?

·        Line 235 – “Cultural f identify” – spelling and grammar mistake

·        Line 235 – “our results show us” is causal language perhaps “results indicate”?

·        Line 236 – “dealing” is used again - has a negative connotation, neutral term is needed (e.g. managed, processed, etc.)

·        Line 241 – “satisfaction.in” spelling error

·        Line 244 – Authors need an introduction part of the sentence, perhaps “Limitations include”

·        Line 246 – “the sample” is an incomplete phrase, perhaps “sample size”?

·        What about future direction of research? Do the authors have any ideas on how to get more data or improve upon the data?

Author Response

Responses to reviewers. Manuscript ID- 2303815

First of all, we would like to thank the reviewers for their suggestions, which we believe were necessary and have helped to improve the quality of this study. All changes have been highlighted in red in the manuscript. We now proceed to respond to each of their suggestions:

REVIEWER 3

  1. References are both in English and Spanish – how is the journal going to handle the ethical and academic responsible for a reader to review the articles? Indeed, we have noted numerous errors in the bibliography. We erroneously included references in Spanish when in fact they were in English. We have also substituted some Spanish references for others that were used in the initial design of this study but were not included in the initial version sent to Healthcare.

  1. If a table is to be created about sociodemographic data all should be included – the table is missing: ages. Table 1 contains variables with N (%), and age M (SD), for this reason it was decided not to include it. Data on age were already included in the text. They are currently in the Results section, as suggested by another reviewer.
  2.  A table needed to show the prevalence rates of obstetric valuable in the same way the authors completed one for sociodemographic data. These data were already shown in tables 2 and 3. They were broken down in this way to organize the results
  3. Researchers fail to state their hypothesis. Although this section is sometimes advised, the journal's instructions were followed. These instructions do not include hypotheses, only the objective of the study.
  4. Authors bring up the point about cultural identity of the midwife and how that can be different than the mothers. This is a valid and crucial point in clinical practice of multiculturalism. However, in their article and research section no further point about the differences between midwife and patient culture is mentioned. It is just brough up once and left alone. Later in the article it discusses cultural identity but refers to the mothers. Authors need to remove the language about differences in culture between midwife & mother or address it more in the article. As it is currently written the differences of culture between midwife and mother is a stand alone irrelevant point. This commentary is provided to contextualise the current reality of childbirth care in Melilla and globally.
  5. Results section can be better organized by dividing it into subtitles of the result. As it is currently written it is one long paragraph and readers can be confused about the points concluded. They have been reorganised
  6. Missing from abstract = brief sentence describing population (e.g. N size, population, etc.). The change has been included
  7.  Line 18 – “dealt with” has a negative connotation, neutral term is needed (e.g. managed, processed, etc.). The change has been included
  8. Authors make it a point to described childbirth as both positive and negative so a neutral term is needed. This research collects information on the satisfaction or dissatisfaction expressed by women. We commented that the experience of childbirth can be positive or negative depending on many factors and that this influences women's satisfaction or dissatisfaction. Neutralising this concept would distort the bibliography consulted
  9. Line 19 – “the objective” incomplete phrase – perhaps “the objective of the study”. The change has been included
  10. Line 21 – “it is” needs to be rephrased to – “this study is” .The change has been included
  11. Line 22 – “was made” – researchers did not make N size, perhaps meant “selected”. The change has been included
  12. Line 23 – “no relationship” is not capitalized which is inconsistent with prior words after headings. The change has been included
  13. Line 25 – “however” is not needed. Suggested change made
  14. Line 26 – “cultural” is not capitalized which is inconsistent with prior words after headings. The change has been included
  15. Line 27 – “Results found that” language is needed. The change has been included
  16. Line 28 is most appropriate for conclusion/discussion section. We have followed the instructions of the journal that indicated Conclusion
  17. Line 32 – The point of this sentences is unclear. Is culture the diversity? This point refers to the diversity of cultures that offer a multitude of ways of coping with or perceiving childbirth.
  18.  Line 37 – “how this is face” – how what is faced: childbirth? Pain? Clarification included in text
  19. Line 41 – incorrect usage of “expressed” – perhaps authors mean “experience”. The authors refer to women's expression, not experience, supported by the bibliography used in this section.
  20. Line 44 – What stage? We change the word into text
  21.  Line 48 – Missing a period. The change has been included
  22. Line 50 – This is not a new paragraph. This is a continuing idea from prior sentences. The change has been included
  23. Line 50-52 – This sentence cannot be understood as it is currently written and has no clear point. Perhaps authors are stating the prevalence of satisfaction is greater when pain is treated appropriately? Are the factors of appropriate pain management include emotional support, pregnancy planning, maternal education? According to the literature, all these factors positively influence the childbirth experience
  24. Line 53 introduces the idea of migration. Authors need a way to clearly divide the introduction with subheadings (childbirth, migration, etc.). The journal's instructions for titles and subtitles have been followed.
  25. Line 55 – “feminisation” is word to describe submissive sexual practices (dominance, submission, kink). More appropriate word choice is needed. This word has been used by other authors when talking about the increase of migrant women, being, in some of the articles used, part of the title. This concept refers to the increase in the number of migrant women. This fact means that Obstetrics and Gynaecology services are increasingly in demand by these women.
  26.  Line 60- “greatly appreciate” – women do not appreciate multiculturalism in this context. Are authors trying to state the women in the childbirth process have benefits because of the intersections of cultures between midwife & mother to be? With this phrase the authors want to express that, when we work assisting births, we can see the richness of expressions and ways of facing this experience according to the woman's culture.
  27.  Line 63 – “in short” wordiness, not needed. It retracts from authors argument. Suggested change made
  28.  Line 64 – who is “us”? researchers, midwives, doctors, pain doctors, community as whole? Health workers and researchers
  29.  Line 64 – “or not” wordiness, not needed. The change has been made
  30.  Line 65 – “alleviate it” needs to be clearly written because it is an unclear reference, alleviate pain? This phrase speaks of pain
  31. Line 66 – “or whether, in contrast, culturally they usually give birth alone” wordiness. Authors use the phrase “acceptance of companionship” which already insinuated labor alone or with company. In the city where the study was conducted, many cultures coexist. We observed that many women refuse to be accompanied during childbirth because that is what their culture dictates.
  32. Line 63-67 – perhaps authors can benefit from the phrase “birth preferences/birth plan of mother influences by her culture?” We believe that the new wording has helped to clarify this idea.
  33. Line 67 – “we” who is we referring to? Health workers and researchers
  34. Line 67 – 69 – confusing sentence. Are these facts or preferences selected by the mother due to culture? Authors are listing examples of these – exaggerated verbalization, no contact with male partner, continue praying? This is a fact that has been widely studied and contrasted by authors who also work in delivery rooms. Women belonging to certain cultures tend to behave in a similar way, and this behaviour is very different from that of women from different cultures. This was the reason why the working group started to study the cultural influence on the birth process.
  35.  Line 69 – missing comma. Change not made due to not having detected the precise location
  36. Line 70-72 is not a new paragraph. This is a concluding statement from prior paragraph. The change has been made
  37.  Line 71-72 – do authors mean “allows the reduction of dissatisfaction of parturient women during labor”? Yes
  38. Primary languages spoken by mothers is missing (percentages of what mother spoke what language) in tables. This is attempted to be address in line 77-81 but authors are not clear. The exact languages spoken by the women were not recorded, as the only data that influenced delivery care was whether or not they had a language barrier. In the paragraph, we tried to contextualise the city in order to provide more knowledge about it. It was not the authors' intention to know specifically the language spoken, only whether there was a language barrie
  39.  Line 77-81: Authors do not need to justify the WHY behind the different languages spoken in the area. This sentence can be removed. Just the data is needed. We thought to contextualise the linguistic characteristics of the area so that the reader would understand the existing problems in the city regarding the language barrier.
  40.  Line 73 – what is the name of the border city? What do locals call the location? How is it referred to? The change has been made
  41.  Line 94 – can be moved to line 92 to make sentence clearer – perhaps obstetric values from last childbirth experience The change has been made
  42.  Line 95 – incorrect tense – perhaps authors mean “experience” The change has been made
  43.  Line 98 – no comma needed The change has been made
  44.  Line 127 – incorrect spelling “es-tab-lished” The change has been made
  45. Line 129 – incorrect spelling “question-naire” The change has been made
  46. Line 127 – 130 – sentence needs to be restricted to clearly state inclusion and exclusion criteria. One sentence for inclusion. One sentence for exclusion The change has been made
  47.  Line 130 – why were inconsistence answers excluded We obtained numerous "illogical" answers such as: age at childbirth 70
  48. This could be a major design flaw biasing the data. Researchers need to give a brief rationale for excluding inconsistence and what was the criteria for inconsistencies We believe that the mistake was to use the word "inconsistencies" as we wanted to express errors in the completion of the questionnaire. It has been removed
  49. Line 146 -  “firstly” is not needed. This is a study, researchers do not need to show step by step how analysis was completed. The change has been made
  50.  Line 146 – who is “it”? We refer to epidural anaesthesia.
  51.  Line 150 – A briefer sentence of table 2 needed. If readers want more information they can seek out table 2 The change has been made
  52.  Line 148-149 – this is repeated from line 132-133. Authors need to list results not repeat what test was selected. The change has been made
  53.  Line 157 – “p =.05” it is odd that researchers are using an “=” sign, perhaps “p<.05”? The change has been made
  54.  Line 159 – “which explains this result” – explains what result? The change has been made
  55.  Line 159 – Incorrect grammar, perhaps “no significant differences WERE observed” The change has been made
  56. Line 160 – the method of Bobath needs to be a separate sentence The change has been made
  57. Line 171 – “the results itemised in Table 3” language awkward, perhaps authors mean “see table 3” The change has been made
  58.  Line 174 – t is lowercase, needs to capitalized The change has been made
  59.  Table 4 – there is a comma in Agnostic (4,09) The change has been made
  60.  Line 200 – “to this end” is casual filler language, can be removed The change has been made
  61.  Line 202 – “reveals” needs to be passed tense The change has been made
  62.  Line 206-209 can be combined with prior paragraph The change has been made
  63.  Line 206-207 – incorrect grammar, perhaps “it was observed” The change has been made
  64.   Line 206-209 – unclear what authors are trying to state regarding non-existence and motive. Authors did not examine motive. The results of the reasons for not accepting an epidural are shown in Table 2.
  65.  Line 210 – “accompaniment during childbirth is an important factor for midwives”. This is a brand-new idea and it is unclear where this sentence is substantiated from the data. Perhaps authors mean it is important for mothers, not midwives? The change has been made
  66.  Line 212 – how is Arbues Cultural identity decisive when choose a companion? Did researcher time mothers on how long it took them to choose a partner? Perhaps decisive is not the appropriate word choice? The word Arbues should not be in the text. It has been removed, giving more meaning to the sentence.·     
  67. Line 215 – 216 – this sentence is more appropriate for introduction section as it discusses prior literature. We believe it reinforces the understanding of the discussion
  68.    Line 218 – “this fact” fact is not the appropriate word choice, this is a preference by mothers not fact. Perhaps researchers mean observation or preference? The change has been made

  1. Line 222 – use of the word “fact” - fact is not the appropriate word choice, Perhaps researchers mean observation or preference? The change has been made
  2. Line 225 – “according to these authors” which authors – the authors of this study or Fernandez-Carrasco et. Al? We refer to the authors quoted in this section
  3.  Line 235 – “Cultural f identify” – spelling and grammar mistake The change has been made
  4.  Line 235 – “our results show us” is causal language perhaps “results indicate”? The change has been made
  5.  Line 236 – “dealing” is used again - has a negative connotation, neutral term is needed (e.g. managed, processed, etc.) The change has been made
  6.  Line 241 – “satisfaction.in” spelling error The change has been made
  7.  Line 244 – Authors need an introduction part of the sentence, perhaps “Limitations include” The change has been made
  8. Line 246 – “the sample” is an incomplete phrase, perhaps “sample size”? The change has been made
  9.  What about future direction of research? Do the authors have any ideas on how to get more data or improve upon the data? The change has been made

In the hope that the amendments will be considered positively by the reviewers, we send our warmest regards